# Fabrication of Ultra-Fine Micro-Vias in Non-Photosensitive Polyimide for High-Density Vertical Interconnects

**DOI:** 10.3390/mi13122081

**Published:** 2022-11-26

**Authors:** Yao Wang, Chuan Hu, Xun Xiang, Wei Zheng, Zhendong Yin, Yinhua Cui

**Affiliations:** 1Institute of Semiconductors, Guangdong Academy of Sciences, Guangzhou 510650, China; 2School of Engineering and Technology, China University of Geosciences, Beijing 100190, China; 3National Engineering Laboratory of Modern Materials Surface Engineering Technology, Guangdong Provincial Key Laboratory of Modern Surface Engineering Technology, Institute of New Materials, Guangdong Academy of Sciences, Guangzhou 510651, China

**Keywords:** polyimide, inductively coupled plasma, etching rate, micro-vias

## Abstract

With the growing demands for transferring large amounts of data between components in a package, it is required for advanced packaging technologies to form smaller vertical vias in the insulators. Plasma etching is one of the most widely used micro-vias formation processes. This paper has developed a fabrication process for 5–10 µm residue-free micro-vias with 70° tapered angle in polyimide film based on O_2_/CHF_3_ inductively coupled plasma (ICP). The etch rate would monotonically increase with the ICP power, RF power, and gas flow rate. As for the gas ratio, there is an optimum range of CHF_3_ ratio, which could obtain the highest etch rate. The results have clearly shown that the enhancement of ion bombardment and prolongation of etching time would be beneficial to grass-like residue removal. In addition, during the etching of partially cured polyimide, the lateral etch rate would significantly increase in the region near the metal hard mask.

## 1. Introduction 

Polyimide (PI) is a widely used insulating material in integrated circuits packages because of its outstanding properties, such as superior dielectric properties, excellent chemical and thermal stability, and high mechanical strength. Although direct chip stacking in 3D integration with silicon vias provide the highest bandwidth between chips [1], thermal and yield challenges have forced the adoption of side-by-side integration with high-density input-output (I/O) interconnections. This requires advanced packaging technologies with increasingly smaller vertical via in polyimide to satisfy the growing demand for a very high I/O count [2].

The major vertical micro-vias formation processes in package technologies are photolithography, laser ablation [3], and plasma etching [4,5]. In the photolithography process, photosensitive dielectrics could form all the micro-vias on the panel or wafer level at once. With photosensitive dielectrics, ultra-small micro-vias below 10 μm [6] could be obtained. However, there are some drawbacks, such as a limited selection of materials and inferior material properties due to the requirement of photosensitivity. From this aspect, laser ablation and the plasma etching process would have a better universality. The major difference between the above two processes is the process property, which means that laser ablation is a sequential process, and the manufacturing process time increases drastically when several millions of vias need to be achieved [2]. 

As a widely used micro-vias formation process, plasma etching of polyimides has been extensively explored. During the research of sidewall profile in plasma etching of polyimide, Deschler et al. [7] investigated the influence of O_2_/CF_4_ gas composition on the sidewall profile of polyimide layers with photoresist as a mask. Zawierta et al. [8] conducted a detailed investigation that demonstrated that the sidewall profile can be controlled by varying the chamber pressure during the etching process. As for the residues issue during polyimide etching, Bliznetsov et al. [9] considered different gas compositions to avoid the generation of residues. In addition, Ma et al. [10] demonstrated the influence of the bias plate power on the solution of residues during etching in pure O_2_ plasma. With regard to the fabrication of micro-vias in polyimide film, Mimoun et al. [11] presented a two-step etching recipe combining both isotropic and anisotropic profiles, resulting in wine-glass-shaped vias with a diameter of 5–10 µm.

In this work, the plasma etching process of non-photosensitive polyimide is exploited to develop a one-step fabrication process for ultra-fine micro-vias. Firstly, the etching mechanism of polyimide is elaborated in detail. Then, the correlations of etching rate with ICP power, RF power, gas flow rate, and ratio are investigated, and the principles within them are well explained. During the investigation, the metal hard mask crack issue is solved by adjusting the elemental composition. Next, the different etching behaviors between partially cured PI and fully cured PI are illustrated. Finally, the residue-free micro-vias with a diameter from 5 to 10 µm are fabricated based on anisotropic etching of polyimide layer under O_2_/CHF_3_ plasma.

## 2. Materials and Methods

### 2.1. Experimental Process

Figure 1 shows a schematic view of the process flow for the fabrication of micro-vias. Briefly, the non-photosensitive polyimide (PI) precursor was spun-coated at 1000 RPM for 60 s on a 2-inch silicon dioxide (SiO_2_) wafer and fully cured under 300 °C in a nitrogen environment for an hour to achieve a 4–5 μm thickness film (Figure 1a). Then, a positive photoresist (PR, AZ4620, Microchem, Guangzhou, China) was spun-coated on the fully cured polyimide at 4000 RPM for 30 s to achieve 4-μm thickness. Next, the photoresist was patterned under UV light and developed in AZ 300 MIF developer (Microchem) for 2 min 30 s. Sequentially, the Ti(20 nm)/Cu(380 nm)/Cr(30 nm) layers were deposited on the patterned sample by e-beam evaporation and the residual photoresist was lifted off by being immersed in acetone for 5–10 min. As a result, hundreds of sample cells (Figure 2) were fabricated on the wafer. As shown in Figure 2, the sample cell consists of three parts. The left and right parts were exactly the same and consisted of metal mask with designed 1, 2, 3, 4, 7, and 9 µm diameter holes. The middle part consisted of metal mask with 80 µm diameter holes.

Finally, the plasma etching of polyimide was carried out in a commercial inductively coupled plasma (ICP) reactor PlasmaPro System 133 ICP380 (Oxford Instruments, Guangzhou, China) with O_2_/CHF_3_ mixture gas. Although numerous investigations suggest the addition of CF4, Nomin Lim et al. [12] considered CHF_3_ could provide much higher etching anisotropy, we finally chose the addition of CHF_3_. The plasma was generated by a 13.56-MHz top electrode power. The bottom electrode power was supplied by RF generator to induce the DC self-bias voltage. During the etching process, the chamber pressure was kept at 20 mTorr.

### 2.2. Analysis

The etch profiles were characterized using a focused ion beam-scanning electron microscope (FIB-SEM, Zeiss Auriga Compack, Guangzhou, China). Etch rate of every sample was determined in the middle part of the sample cell through the mechanical measurement using a DektakXT^®^ stylus profilometer with 2 µm stylus tip radius.

### 2.3. Etching Mechanism

According to numerous studies on the mechanism of polyimide etching, benzene rings are one of the stable components in polyimide films [13]. Thus, in the plasma etching process of polyimide, etch gases and etch gas mixtures accounted for mostly pure O_2_ or combinations of O_2_ and CF4/SF6/CHF_3_ [4,9,14,15]. During the process, the main etchants of the polyimide are oxygen and fluorine atoms.

In the plasma etching of polyimide under pure oxygen, it is assumed that the carbon atoms of benzene rings in the polyimide structure are oxidized by atomic oxygen [11]. Hence, the etch rate depends mainly on the concentration of atomic oxygen. However, it has been already reported that without charged species and photo radiation, the etching of polyimide will be negligible in pure oxygen plasma [16]. Therefore, the etching mechanism of polyimide in oxygen and fluorine-containing gas could be suspected as follows:Plasma ion bombardment on the polyimide surface could create reactive sites;The addition of small amounts of fluorine-containing gas could greatly enhance the production of atomic oxygen [11];Fluorine atoms could react with unsaturated groups in the polyimide to produce a saturated intermediate polymer [4,17] and reduce the etching activation energy [15];Hydrogen atoms in the polyimide structure could be abstracted by fluorine to generate HF molecules and surface radical sites;The surface radical sites could easily bond with oxygen atoms and fluorine atoms;The bond between radical sites and oxygen atoms [11] would decrease the strength of carbon-carbon bonds allowing the carbon oxidation in benzene rings and the destruction of polyimide molecule structure [17];The bond between radical site and fluorine atoms would form a stable fluorinated surface.

## 3. Results and Discussion

### 3.1. Etch Rate vs. ICP Power

A comparable set of experiments was conducted to relate etch rate and ICP power under constant RF power (500 W), etch gas composition (CHF_3_ 10%), and etch gas flow speed (100 sccm). The correlation of etch rate on ICP power is depicted in Figure 3. It is shown that the maximum etch rate of 2.36 μm/min could be obtained under 3000 W ICP power. In the etching process of the polyimide, increasing ICP power substantially enhances the etch rate. However, it has been found that the metal mask would have a chance to crack under an ICP power of more than 2000 W.

With the increasing ICP power, the density of ions, atoms, and radicals in the plasma increased. According to the above suspected etching mechanism, the increased density of ions would strengthen the ion bombardment. The increased fluorine atoms could generate more saturated intermediate from the unsaturated groups in the polyimide and abstract more hydrogen atoms. The increased oxygen atoms could react with more radical sites. This means more reactive sites generated in the polyimide structure and the enhancement of physical and chemical reactions between fluorine/oxygen atoms and the reactive sites, resulting in a positive effect on the etching rate. However, there was an upper limit of ionization degree with the increasing ICP power. Thus, the gain of etch rate decreased with the increasing ICP power.

### 3.2. Etch Rate vs. Gas Composition

Figure 4 depicts the resulting etch rates when varying etch gas composition while keeping ICP power (2500 W), RF power (500 W), and gas flow speed (100 sccm) constant. The highest average etch rate of approximately 2.3 μm/min was obtained within the CHF_3_/(O_2_ + CHF_3_) ratio range from 10% to 20%, and the etch rate was dramatically decreased when the CHF_3_ gas flow ratio increased to 30%.

According to the etching mechanism, the increased fluorine atoms in the plasma would induce more oxygen atoms and enhance the abstraction of hydrogen in the polyimide film and surface radical sites amounts in favor of the production of intermediate polymer and reduction of the etching activation energy [13]. As a result, the etch rate would increase with the increased CHF_3_ ratio. However, during the etching process, fluorine atoms could also bond with the radical sites to form a stable fluorinated surface. The bonding of excess fluorine atoms with polyimide molecules would passivate the film surface, resulting in the reduction of etch rate for continually increasing the CHF_3_ ratio.

### 3.3. Etch Rate vs. RF Power

The aim of the experiment was to avoid the crack issue of the metal mask when the ICP power is greater than 2000 W. The etch rate as function of RF power is shown in Figure 5 while the ICP power, gas flow ratio, and speed are fixed at 2000 W, 10% and 100 sccm. The etch rate increased with increasing RF power, but the growth rate decreased. Under 600 W RF power, a maximum etching rate of 2.23 μm/min was obtained.

When the RF power increases, the ion energy during bombardment increases [18], resulting in more reactive sites on the polyimide surface. Therefore, the bonding amounts between these reactive sites and oxygen/fluorine atoms were enhanced, and the etch rate was increased.

### 3.4. Etch Rate vs. Gas Flow Speed

Figure 6 shows the etch rate versus the total gas flow rate with 2000 W ICP power, 600 W RF power, and a CHF_3_/(CHF_3_ + O_2_) gas ratio of 10%. A nearly linear dependence of etch rate on etch gas flow speed was found. The results show that the etch rate increases with increasing gas flow speed, and an etch rate of 2.42 μm/min was obtained.

According to Buder [15], the increasing total gas flow speed would reduce the residence time of particles in the chamber and constantly add uncharged species to the plasma, thus, reducing the effectiveness of ion bombardment to generate reactive sites. Therefore, increased gas flow speed corresponds to lower etch rates. In this case, although gas flow speed was increased in the range from 50 to 150 sccm, the residence time was enough for ions, radicals, oxygen, and fluorine atoms to reach the polyimide surface and chemically or physically react with the PI molecule structures. At the same time, increased gas flow speed would supplement the active species and ion fluxes in the plasma [4], causing the increased etch rate.

### 3.5. Hard Mask Crack Issue

During the research, a major challenge was the crack issue of the metal hard mask after the residual lift-off process (Figure 1e). As shown in Figure 7, the hard mask on the fully cured PI film is metal Cr film with a thickness of 100 nm. It clearly shows the crack of the Cr mask and the resulting delamination between the mask and PI film. 

This crack issue is attributed to the characteristic of metal Cr and the fabrication process of the samples. Firstly, the metal Cr film is very fragile. Then, during the residual PR lift-off process, the sample was immersed in acetone for 5 min. Though the PI film had already been fully cured under 300 °C for an hour, the acetone molecules would still be absorbed into the interspace between the backbones of the PI film, causing the very slight swelling of the polyimide structure. Finally, the cracking of the Cr mask happened.

The above problem was solved by adjusting the composition of the metal hard mask. When the Ti/Cu/Cr multiple metal layers (thickness of 20, 380, and 30 nm, respectively) were deposited as a hard mask, due to the high ductility of copper, the integrality of the hard mask would not be broken after the lift-off of the residual PR, as shown in Figure 8, Figure 9, Figure 10 and Figure 11.

### 3.6. Partially Cured Polyimide

During the curing of polyimide, the PI precursor film began to partially imidize around 180 °C and was fully cured at around 300 °C [13]. Figure 8 shows the etching results of the partially cured PI film, which was just cured under 180 °C for three hours. Figure 9 shows the etching results of the fully cured PI film under the same etching conditions within Figure 8. From the aspect of vertical etching, it is shown that the etch rate of partially cured PI is slightly slower than the fully cured PI, which is completely opposite from the results of Azeem Zulfiqar’s [19]. From the aspect of lateral etching, it is clearly shown that the length of the undercut of the partially cured PI is significantly longer than that of the fully cured PI film. For the micro-vias with a diameter of nearly 4 μm, the undercut lengths are 2.5 μm for partially cured PI. 

For the fully cured PI, the undercut length is just 1.5 μm. As for the micro-via with a diameter of 9 μm, the length for partially cured PI is 2.5–2.9 μm and for fully cured PI is nearly 2 μm. Most importantly, as shown in Figure 8, the sudden increase of the undercut is mainly located in the region near the hard mask.

It has been reported that the lateral thermal conductivity of spin-coated polyimide film is larger than the vertical thermal conductivity [20,21,22] due to the shear stress in spin coating and the ordering of the molecular chains in the surrounding matrix. In other words, during the curing process of polyimide, the orientation of the formed large molecule chains is more likely parallel with the substrate. The strong coupling of atomic vibrations along the chains enhances energy transport, resulting in a higher thermal conductivity. The weak Van der Waals interaction between the neighboring chains impedes the transport of atomic vibrational energy [20].

As for the partially cured PI film, the fraction of formed large molecule chains whose orientation is parallel with the substrate is much smaller than the fully cured PI, which means the lateral plasma etching of partially cured PI would be easier than the fully cured PI film. At the same time, in the etching process, the heat of the hard mask because of the ion bombardment and plasma heating could not be effectively transferred to the substrate [8]. Thus, the temperature of the metal hard mask was higher than the underneath PI film. Higher temperature would have a positive effect on the chemical reactions, resulting in a faster etching rate around the interface of PI film and hard mask [13]. From this point, the above phenomena may be explained by the fraction of large molecule chains and differences in thermal conductivity. In the fully cured polyimide structure, the complete polymerization and more ordered arrangement of the structure resulted in more molecule chains and higher thermal conductivity. Thus, in the partially cured PI, the temperature of the interface between metal mask and PI film was higher than that of the fully cured sample, but the fraction of polyimide molecule chains parallel with the substrate was less. As a result, enhancing the etching rate of the region.

### 3.7. Residue Issue

From the correlation of etching rate with ICP power, RF power, gas flow ratio, and flow rate, Figure 9 and Figure 10 show the FIB-SEM pictures of fully cured PI micro-vias etched under the conditions of 2500 W ICP power, 500 W RF power, 10% CHF_3_ ratio, 100 sccm gas flow and 2–3 min.

In Figure 9, the grass-like residues in the micro-via are observed on the substrate. According to the above-mentioned etching mechanism, there are mainly two reasons for the etching residues: The inability of ion bombardment to generate sufficient reactive sites, the absence of fluorine atoms [19] in the plasma, which could produce a saturated intermediate polymer, surface radical sites, and reduce the etching activation energy. During the etching process, the by-products of the reaction between oxygen atoms and polyimide [10] would hinder the continual reaction. Under this situation, the ion bombardment should be strong enough to generate reactive sites to successfully destroy all polyimide chains. At the same time, prolongation of the etching process [8] would be beneficial to the reaction of the residue with the active species in the O_2_/CHF_3_ plasma. Though the etching process time was prolonged to 3 min, as shown in Figure 10, the grass-like residues were only slightly reduced. However, at the same time, the undercut length was increased by nearly 0.5 μm due to the prolongation of the etching time.

Then, the RF power was increased to 600 W with the fixed gas ratio and rate shown in Figure 9 and Figure 10. In order to reduce the thermal effects on the hard mask, the ICP power was decreased to 2000 W. The results are shown in Figure 11. The length of the undercut was reduced, as expected increased RF power improved the etching anisotropy due to higher ion energy and thus improved the directionality of the etching [18]. However, there were still grass-like residues.

Finally, by prolonging the etching time to 3 min under the same conditions as shown in Figure 11, the grass-like residues were completely cleared for the 9 μm micro-vias, as shown in Figure 12. Because of the inevitability of undercut due to the lateral etching, for the 9 μm holes in the hard mask, the micro-vias with 13 μm upper aperture and 10 μm bottom aperture were fabricated. For the 4 μm holes in the hard mask, micro-vias with 8 μm upper aperture and 5 μm bottom aperture were fabricated. In addition, according to Figure 9, Figure 10, Figure 11 and Figure 12, it is clearly shown that a nearly 70° taper angle would be obtained during the etching process.

## 4. Conclusions

For the formation of ultra-fine micro-vias in non-photosensitive polyimide film, a fabrication process has been developed based on O_2_/CHF_3_ inductively coupled plasma (ICP). Under 2000 W ICP power, 600 W RF power, 10% CHF_3_ ratio, and 100 sccm gas rate, the 5–10 μm diameter micro-vias with 70° taper angle could be obtained with an etching rate of 2.2 μm/min.

The etch rate would increase with the increased ICP power, RF power, and gas flow rate. As for the gas ratio, there is an optimum range of CHF_3_ ratio, which could obtain the highest etch rate.

During the fabrication of the samples, the metal Cr hard mask would crack due to the swelling of cured PI and the fragile characteristic of Cr. The integrality of the hard mask would be kept with the addition of Cu layer.

Compared with the etching of fully cured polyimide, for the etching of partially cured PI, the lateral etch rate would significantly increase in the region near the metal hard mask, due to the poorer thermal conductivity and less polyimide large molecule chains, which are parallel with the substrate. As for the grass-like residues, the results have clearly shown that the enhancement of ion bombardment and prolongation of etching time would be beneficial to the grass-like residue removal.

## Figures and Tables

**Figure 1 micromachines-13-02081-f001:**
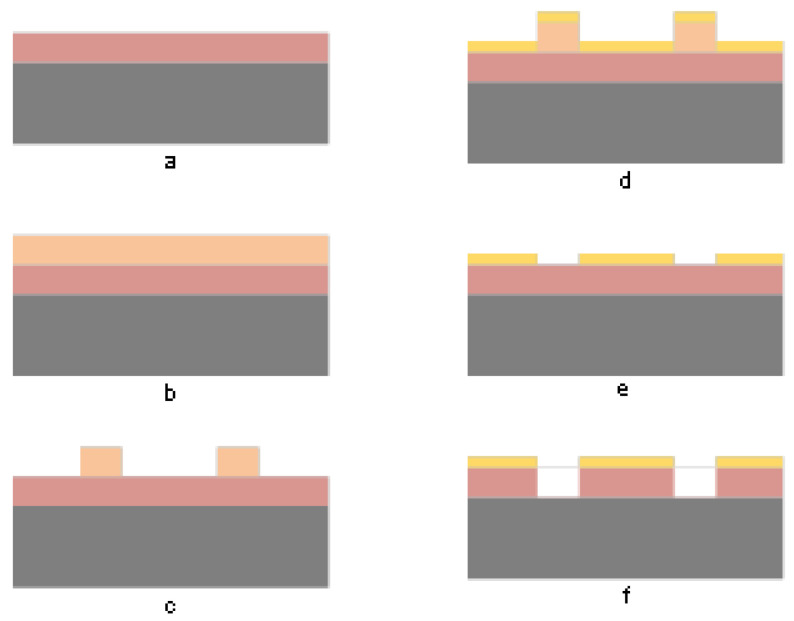
The micro-vias fabrication process: (**a**) PI spun-coated on SiO_2_ wafer; (**b**) PR spun-coated on PI film; (**c**) lithography and development of PR; (**d**) Ti/Cu/Cr hard mask deposition; (**e**) residual PR lifted-off; (**f**) ICP plasma dry etching of PI.

**Figure 2 micromachines-13-02081-f002:**
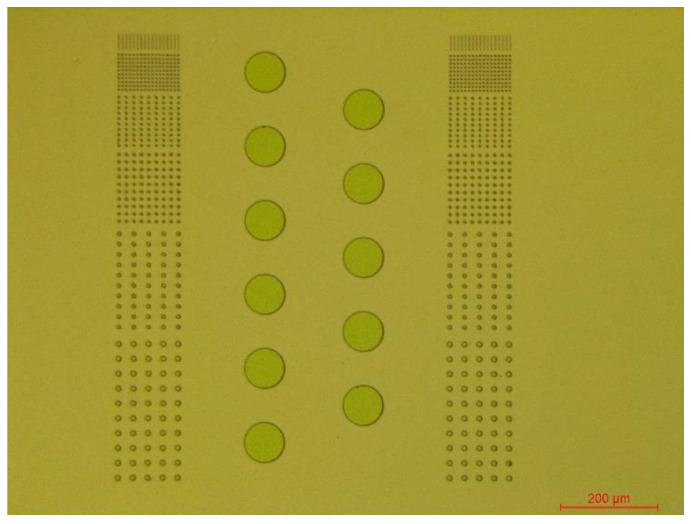
The pattern of sample cell.

**Figure 3 micromachines-13-02081-f003:**
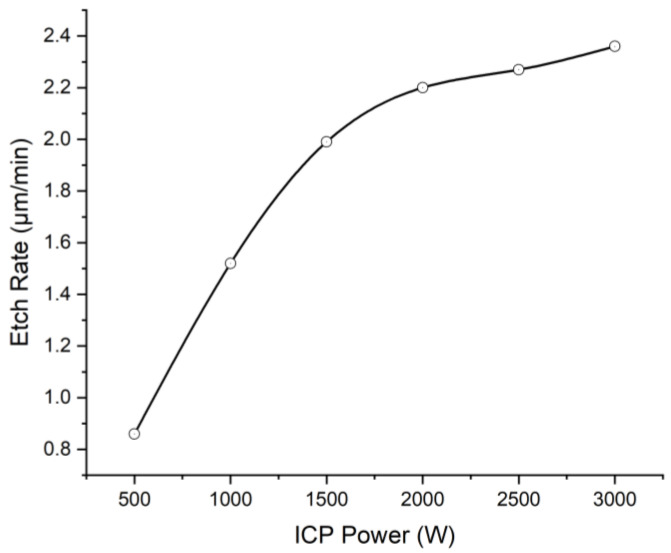
The relationship between etch rate and ICP power under 500 W RF power, 10% CHF_3_ gas composition, and 100 sccm etch gas flow speed.

**Figure 4 micromachines-13-02081-f004:**
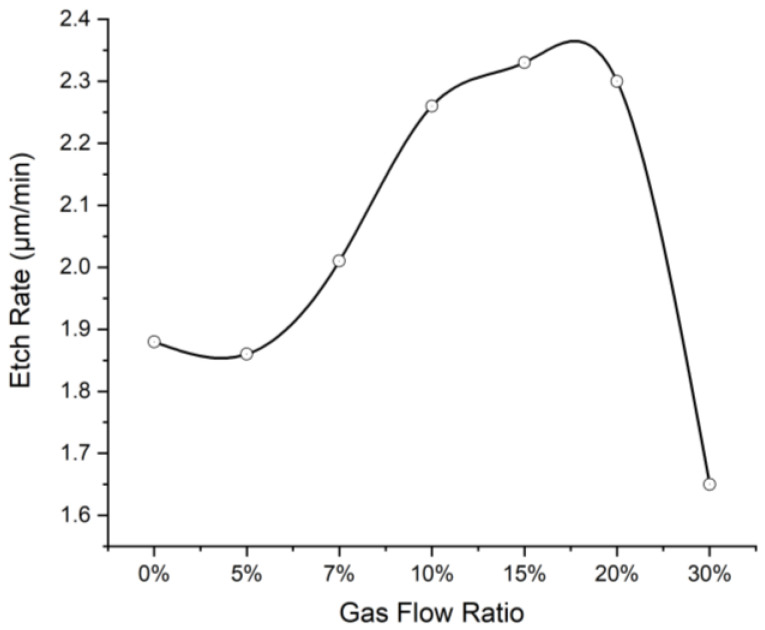
The relationship between etch rate and gas composition under 2500 W ICP power, 500 W RF power, and 100 sccm etch gas flow speed.

**Figure 5 micromachines-13-02081-f005:**
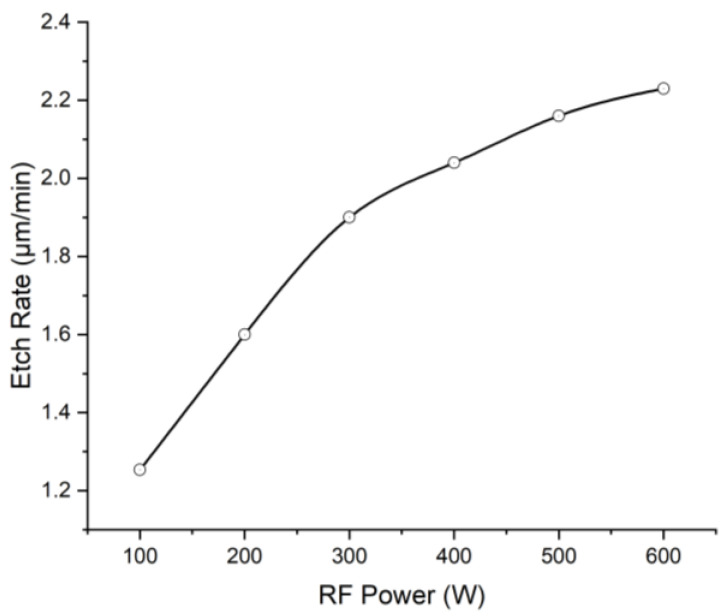
The relationship between etch rate and RF power under 2000 W ICP power, 10% CHF_3_ gas composition, and 100 sccm etch gas flow speed.

**Figure 6 micromachines-13-02081-f006:**
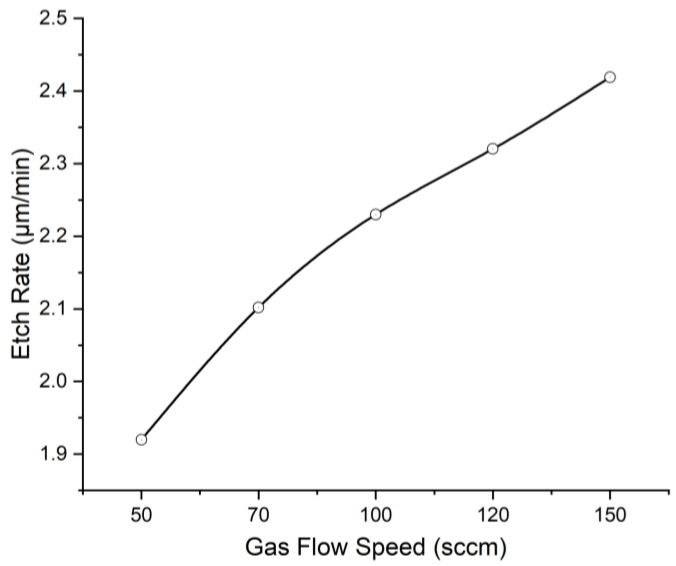
The relationship between etch rate and gas flow speed under 2000 W ICP power, 600 W RF power, and 10% CHF_3_ gas composition.

**Figure 7 micromachines-13-02081-f007:**
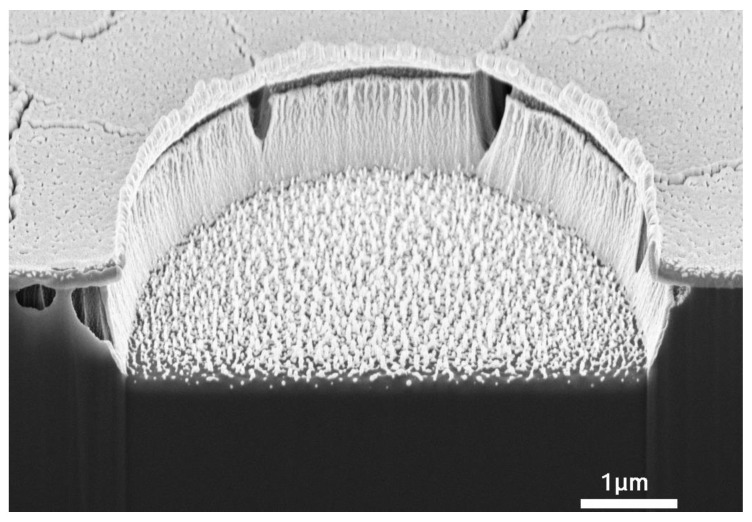
ICP etching of fully cured PI under 2500 W ICP power, 500 W RF power, 7% CHF_3_ ratio, 100 sccm gas flow, and 0.5 min.

**Figure 8 micromachines-13-02081-f008:**
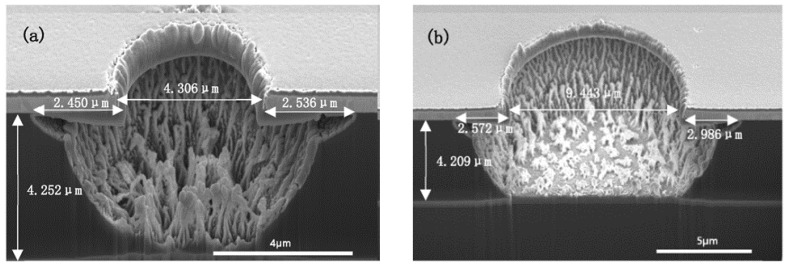
ICP etching of partially cured PI under 2500 W ICP power, 500 W RF power, 10% CHF_3_ ratio, 100 sccm gas flow and 2 min. ((**a**) the via diameter is 4 μm; (**b**) the via diameter is 9 μm).

**Figure 9 micromachines-13-02081-f009:**
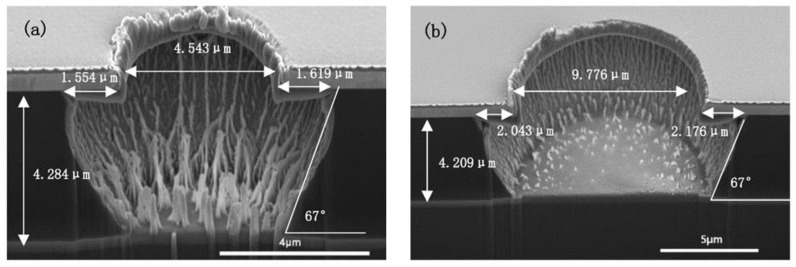
ICP etching of fully cured PI under 2500 W ICP power, 500 W RF power, 10% CHF_3_ ratio, 100 sccm gas flow and 2 min. ((**a**) the via diameter is 4 μm; (**b**) the via diameter is 9 μm).

**Figure 10 micromachines-13-02081-f010:**
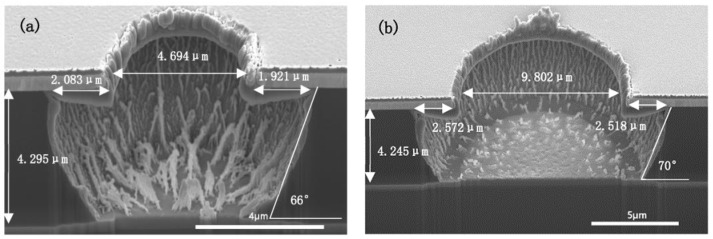
ICP etching of fully cured PI under 2500 W ICP power, 500 W RF power, 10% CHF_3_ ratio, 100 sccm gas flow and 3 min. ((**a**) the via diameter is 4 μm; (**b**) the via diameter is 9 μm).

**Figure 11 micromachines-13-02081-f011:**
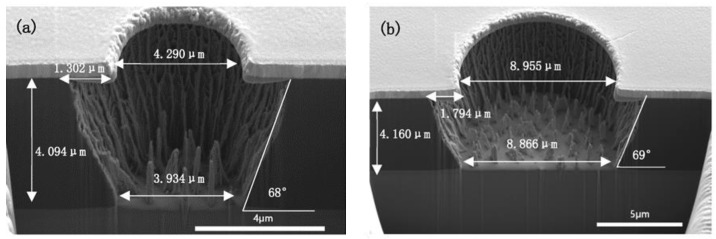
ICP etching of fully cured PI under 2000 W ICP power, 600 W RF power, 10% CHF_3_ ratio, 100 sccm gas flow and 2 min. ((**a**) the via diameter is 4 μm; (**b**) the via diameter is 9 μm).

**Figure 12 micromachines-13-02081-f012:**
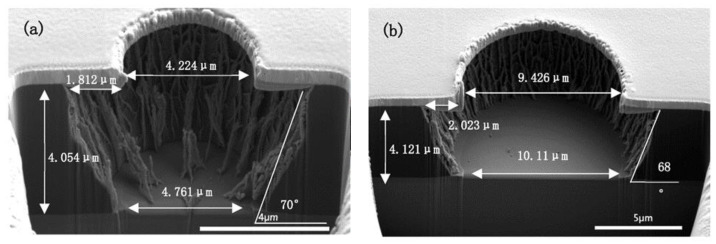
ICP etching of fully cured PI under 2000 W ICP power, 600 W RF power, 10% CHF_3_ ratio, 100 sccm gas flow and 3 min. ((**a**) the via diameter is 4 μm; (**b**) the via diameter is 9 μm).

## Data Availability

This study did not report any data before.

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
