# Peer review of "Fabrication of Ultra-Fine Micro-Vias in Non-Photosensitive Polyimide for High-Density Vertical Interconnects"

_micromachines, 2022, doi:10.3390/mi13122081_

Round 1
Reviewer 1 Report
The manuscript, titled “Fabrication of ultra-fine micro-vias in non-photosensitive polyimide for high-density vertical interconnects” presents dry etching of polyimide layers using oxygen and fluorine-based plasma. The etch rate has been investigated as a function of etching parameters such as power, gas composition and flow rate speed.
I had reviewed this paper for another journal. Unfortunately, the authors have not revised their work according to the reviewers' comments. My comments have not addressed in this version either.
The paper adds marginal information compared to what has already been reported in the literature. I have some issues that have to be addressed before considering the paper for publications.
· The introduction is very weak. Most of its contents is not relevant to the paper topic.
· Since the paper subject is etching, the authors should provide sufficient background about the etching processes of polyimide. There are many studies on polyimide etching. The authors have to mention the most relevant research that have done in the literature.
· Also, the novelty and the message of the paper is missing. Why this work has been done? And how this work is different from previous ones? Why the etching has been done using this plasma in particular? How about other etching options such as oxygen etching?
· How the etch rate has been determined? What is the Profile-system?
· The mechanism of polyimide etching in oxygen- and fluorine-based plasma should be moved in a separate section and linked to the contents of introduction and discussion sections.
· I cannot see the purpose of having tables and figures together explaining the same results. The figures should be enough as one parameter is changing while fixing other parameters.
· There is no consistency in choosing the value of the parameters. For example, they have set ICP power of 2500 W and RF power of 600 W in table 3, then they have set them to 2000 W and 500 W in table 4?! This must be explained otherwise it is just a random selection of values.
· Although the PI has been cured, the deposited Cu metal is cracked. How about the effect of deposition temperature of the metal? Also, I think the curing temperature/ and time of polyimide play a role in the cracking/adhesion issues of the deposited metals as can be seen in figures 7 and 8.
Reviewer 2 Report
The authors report their work on the fabrication of micro-via structures in polyimide. The research motivation was clearly stated, the design of the experiment is systematic, and the experimental work was rigorously done. The topic is relevant to the Micromachines’s readers, therefore the referee recommends the manuscript be published after the following concerns are addressed.
1. In the abstract, the authors wrote that ‘plasma etching is one of the potential micro-vias formation process’, but the plasma etching method is not just a ‘potential’ method, but a most widely-used one.
2. In the abstract, the authors claim their process is ‘residual-free’, but according to the SEM images, it’s not true. ‘Grassing’ defect exists in all these SEM images.
3. In the abstract, the authors wrote that ‘the lateral etch rate would significantly increase in the region near the metal hard mask’, and this is a commonly seen issue in plasma etching called ‘undercut’. The authors later made a reasonable explanation later in the manuscript, but it will be good if the authors can make a comparison with the ‘undercut’ in Si/SiO2 etching.
4. Anycroms need to be defined when they first appear in the paper, not later (e.g. PI).
5. ‘dc self-bias voltage’, use capital letters for ‘DC’.
6. The etch rate was measured with a stylus tip that has a radius of 2 μm, and the measurement can be inaccurate when the diameter of the micro-via opening is small and the aspect ratio of the miro-vias becomes too high. The authors should clarify the angle of the stylus tip, and comment on whether they had problems measuring the deep micro-vias (samples in Fig. 7/8/9/10/11).
7. It is not rigorous to say that ‘the polyimide will not be etched in pure oxygen plasma. The atomic oxygen is not capable of attacking undamaged polyimide structures.’ Actually, given the time, the polyimide layer can be removed physically with plasma generated with any kind of gas. Please revise the statement here.
8. The red measurement labels in Fig. 6/10/11 are barely visible. Please consider removing or changing the color.
9. The authors proposed a tri-layer hardmask to deal with the hardmask cracking issue. How, or is it possible to remove the hardmask after the micro-vias have been etched?
Round 2
Reviewer 1 Report
The authors have addressed my comments in the revised version. However, the responses to my comments (#2 and #6) mentioned in the response letter should be included in the revised paper.
There are many English-related errors (grammar and typos) that should be corrected during the proofreading phase.
Overall, I am happy with the authors' revision, and therefore the paper can be accepted for publication in Micromachines.
Author Response
Thanks for the reviewer's opinions, these comments are very helpful to improve the quality of our manuscript.
Comment: The authors have addressed my comments in the revised version. However, the responses to my comments (#2 and #6) mentioned in the response letter should be included in the revised paper.
Response: The responses to the comments (#2 and #6) mentioned in the response letter have been integrated in the revised paper ( line 93-95, line 166-167 and line 193 in the "Track Changes" mode).
Comment: There are many English-related errors (grammar and typos) that should be corrected during the proofreading phase.
Response: Some errors have been corrected (line 15, line 26, line 27, line 62, line 65, line 279, line 301 and line 337 in the "Track Changes" mode).
Special thanks to you for your good comments.
We appreciate for Editors/Reviewers’ warm work earnestly and hope that the correction will meet with approval.
Once again, thank you very much for your comments and suggestions
Thank you and best regards.
Yours sincerely.